# Plant-Dominant Low-Protein Diet for Conservative Management of Chronic Kidney Disease

**DOI:** 10.3390/nu12071931

**Published:** 2020-06-29

**Authors:** Kamyar Kalantar-Zadeh, Shivam Joshi, Rebecca Schlueter, Joanne Cooke, Amanda Brown-Tortorici, Meghan Donnelly, Sherry Schulman, Wei-Ling Lau, Connie M. Rhee, Elani Streja, Ekamol Tantisattamo, Antoney J. Ferrey, Ramy Hanna, Joline L.T. Chen, Shaista Malik, Danh V. Nguyen, Susan T. Crowley, Csaba P. Kovesdy

**Affiliations:** 1Department of Medicine, Division of Nephrology Hypertension and Kidney Transplantation, University of California Irvine (UCI), Orange, CA 90286, USA; amandab@uci.edu (A.B.-T.); wllau@uci.edu (W.-L.L.); crhee1@uci.edu (C.M.R.); estreja@uci.edu (E.S.); etantisa@hs.uci.edu (E.T.); ferreya@uci.edu (A.J.F.); ramyh1@uci.edu (R.H.); danhvn1@uci.edu (D.V.N.); 2Tibor Rubin VA Long Beach Healthcare System, Long Beach, CA 90822, USA; jolinec@uci.edu; 3Department of Medicine, New York University Grossman School of Medicine, New York, NY 10016, USA; shivam.joshi@nyulangone.org; 4Lexington VA Healthcare System, Lexington, KY 40502, USA; rebecca.schlueter@va.gov; 5Kansas City VA Medical Center, Kansas City, MO 64128, USA; Joanne.Cooke@va.gov; 6Flavis/Dr. Schar USA, Inc., Lyndhurst, NJ 07071, USA; meghan.donnelly@drschar.com; 7UCI Health Susan Samueli Center Integrative Health Institute, Irvine, CA 92626, USA; sschulm1@hs.uci.edu (S.S.); smalik@hs.uci.edu (S.M.); 8VA Connecticut Healthcare System, West Haven, CT 06516, USA; Susan.Crowley@va.gov; 9Division of Nephrology, Yale University School of Medicine, New Haven, CT 06516, USA; 10Division of Nephrology, University of Tennessee Health Sciences Center, Memphis, TN 38163, USA; ckovesdy@uthsc.edu

**Keywords:** plant-dominant, low-protein, dietary protein intake, glomerular hyperfiltration

## Abstract

Chronic kidney disease (CKD) affects >10% of the adult population. Each year, approximately 120,000 Americans develop end-stage kidney disease and initiate dialysis, which is costly and associated with functional impairments, worse health-related quality of life, and high early-mortality rates, exceeding 20% in the first year. Recent declarations by the World Kidney Day and the U.S. Government Executive Order seek to implement strategies that reduce the burden of kidney failure by slowing CKD progression and controlling uremia without dialysis. Pragmatic dietary interventions may have a role in improving CKD outcomes and preventing or delaying dialysis initiation. Evidence suggests that a patient-centered plant-dominant low-protein diet (PLADO) of 0.6–0.8 g/kg/day composed of >50% plant-based sources, administered by dietitians trained in non-dialysis CKD care, is promising and consistent with the precision nutrition. The scientific premise of the PLADO stems from the observations that high protein diets with high meat intake not only result in higher cardiovascular disease risk but also higher CKD incidence and faster CKD progression due to increased intraglomerular pressure and glomerular hyperfiltration. Meat intake increases production of nitrogenous end-products, worsens uremia, and may increase the risk of constipation with resulting hyperkalemia from the typical low fiber intake. A plant-dominant, fiber-rich, low-protein diet may lead to favorable alterations in the gut microbiome, which can modulate uremic toxin generation and slow CKD progression, along with reducing cardiovascular risk. PLADO is a heart-healthy, safe, flexible, and feasible diet that could be the centerpiece of a conservative and preservative CKD-management strategy that challenges the prevailing dialysis-centered paradigm.

## 1. The Burden of Chronic Kidney Disease

Chronic kidney disease (CKD) has no cure and affects more than 10% of the adult population throughout the world [1]. If persons with CKD survive long enough, many will inevitably reach kidney failure, also known as end-stage kidney disease (ESKD), which is not compatible with life without kidney replacement therapy in the form of maintenance dialysis treatment or kidney transplantation [1,2,3,4]. However, ESKD patients who transition to dialysis often have poor clinical outcomes. Cardiovascular morbidity and mortality of CKD are exceptionally high with an overall five-year survival less than 50% [5,6]. In the United States (US), total Medicare and Veterans Administrations (VA) spending for CKD continues to increase [4]. Each year, approximately 120,000 Americans develop ESKD and initiate dialysis [5], including 12,000 U.S. Veterans [5,6,7,8]. Despite the purported life-prolonging effects of dialysis [9], 10% of these patients die in the first 90 days after dialysis transition and >20% in the first year [6]. In addition to the high rates of early dialysis mortality, a large proportion of elderly patients experience major functional decline after transition to dialysis therapy [10]. Hence, delaying or preventing kidney failure to avoid kidney replacement therapy may improve clinical outcomes while averting the high costs of dialysis therapy and preserving limited resources.

The World Kidney Day steering committee declared 2020 and 2021 as the years of CKD prevention and living well with CKD, respectively. These declarations underscore the paramount importance of both primary CKD prevention as well as secondary and tertiary interventions for early diagnosis of CKD and treatment to control progression to ESKD and its complications, respectively [1,11]. Among the core components of the preventative strategies are nutritional and dietary intervention as featured in this review article. Moreover, in July 2019, an unprecedented Executive Order by the U.S. President, known as the “Advancing American Kidney Health Initiative,” sought to reduce the number of Americans developing kidney failure by 25% by 2030 through improved efforts to slow the progression of CKD [12]. This timely executive order underscores the importance of preventive CKD measures and reiterates the critical, yet underappreciated role of leveraging dietary interventions in optimizing kidney health [12]. This review article highlights past and contemporary data on the dietary management of CKD with focus on the role of plant-based, restricted protein diets based on the premise that feasible dietary approaches should be the cornerstone of non-pharmacologic strategies in slowing CKD progression and avoiding or delaying ESKD [13].

## 2. High Protein Diets May Be Harmful to Kidney Health

While the U.S. National Academy of Medicine has maintained that Recommended Dietary Allowance (RDA) of dietary protein intake (DPI) should be 0.8 g per kilogram of the ideal body weight per day (g/kg/day), Americans on average consume much higher amounts of protein, i.e., 1.2 to 1.4 g/kg/day, mostly from animal sources, according to analyses from the National Health and Nutrition Examination Survey (NHANES) [14]. In recent practice, higher DPI has been recommended to combat obesity and diabetes [15,16], despite recent data suggesting higher risk of CKD incidence and progression with higher DPI, especially from red meat [17,18,19]. Keto-diets, which are also high in protein and animal fats, are gaining popularity across different healthcare systems throughout the world as a recommended dietary intervention for adults with diabetes [20]. Despite its immediate appeal for the use of type 2 diabetes, the ketogenic diet has not been as effective for glycemic control or weight loss in randomized, controlled trials as often touted and may carry additional risks to long-term health [21]. Furthermore, previous and emerging data (Table 1) suggest that high DPI in these diets, by way of causing increased intra-glomerular pressure with resultant glomerular hyperfiltration, may adversely affect kidney health over time across populations with or at-risk for CKD [17].

## 3. A Low Protein Diet Preserves Kidney Function

A low protein diet (LPD), defined as DPI 0.6–0.8 g/kg/day, has consistently been shown to lower intra-glomerular pressure (Figure 1) [25]. This effect, if exerted consistently, may preserve long-term kidney function as corroborated in both animal models and in human studies of CKD, including several meta-analyses [24,26,27,28,29]. The scientific premise for these DPI targets was presented in a recent critical review and meta-analysis of 16 dietary trials with more than 30 CKD patients in each trial (Figure 2) [28], and also discussed in a 2017 *New England Journal of Medicine* review paper [25]. These data highlight the utility of LPD for the management of CKD (Table 2), suggesting that an LPD of 0.6–0.8 g/kg/day vs. higher amounts is associated with lower ESKD risk, higher serum bicarbonate and lower serum phosphorus levels, less azotemia, and lower mortality trends [28]. Whereas we and others have recommended DPI of 0.6–0.8 g/kg/day, some other investigators including Metzger et al. [30] showed that a DPI of <0.6 g/kg/day may result in even slower CKD progression; however, a DPI of 0.6–0.8 g/kg/day is considered the most pragmatic and safest target when used without amino-acid or keto-analogue supplements to avoid protein-energy wasting (PEW). For persons without established CKD but who are at high risk of CKD, such as those with a solitary kidney or diabetic glomerular hyperfiltration, it is recommended that a high dietary protein intake >1.0 g/kg/day should be avoided [31], especially since patients with diabetes develop more severe hyperfiltration in response to high DPI [32].

Evidence suggests that safety and adherence to an LPD is equivalent to a normal protein diet and that there is no risk of the malnutrition or PEW that might occur with very-low protein diets (DPI 0.3–0.6 g/kg/day), even sans supplementation with essential amino acids or their keto-analogues [28]. However, while most studies suggest that an LPD ameliorates CKD progression, there are also some mixed findings [33,34], including the primary analyses of the Modification of Diet in Renal Disease (MDRD) study. Most trials except for the MDRD were small, used surrogate endpoints, were considered less rigorous compared to MDRD, used dietary interventions that were labor-intensive, were not patient-centered, and were not aligned with contemporary culture of more plant-based sources. Due to the impractical aspects of prior LPD regimens, and in part to the marginal effects of an LPD in the MDRD, which did not achieve statistical significance, LPD has not been adopted in most CKD clinics. Thus, there remains an unmet need for more contemporary, well-powered, pragmatic randomized controlled trials that apply LPD as a convenient and patient-centered intervention, especially with a newer focus on plant-dominant diet regimens.

## 4. Plant-Based Foods Have a Favorable Impact on Kidney Health

The typical American diet contains 15–20% protein with less than one-third of protein sources from plants [53]. While human trials on the effects of high protein intake have yield mixed findings, animal models are relatively consistent with evidence of histological damage, including a 60–70% increase in renal and glomerular volumes, 55% more fibrosis, and 30% more glomerulosclerosis [54]. A recent comprehensive and critical review of the literature concluded that daily red meat consumption over years may increase CKD risk, whereas fruit and vegetable proteins may be renal protective [18]. Prior studies summarized by some of the authors of this article [31,33,34,55,56,57,58,59,60,61] suggest that animal-based protein is harmful to kidney health, while a plant-dominant diet may slow CKD progression. A landmark study was presented by Kontessis et al. [62], who studied volunteers fed for 3 weeks with a vegetable-based diet (*N* = 10), an animal protein diet (*N* = 10), or an animal protein diet supplemented with fiber (*N* = 7), all with the same amount of total protein; animal-based protein diets increased GFR more than similar amounts of plant-based proteins, i.e., higher glomerular hyperfiltration was observed with more meat and less vegetable-derived proteins [62]. Other important studies supporting the benefit of a plant-dominant diet in slowing CKD progression include the study by Lin et al. [63] (who examined 3348 women in the Nurses’ Health Study and found that the highest quartile of meat intake was associated with 72% higher risk of microalbuminuria), Kim et al. [64] (who showed that in 14,686 middle-aged adults, higher adherence to a plant-based diet was associated with favorable kidney outcomes), Haring et al. [65] (who showed that red and processed meat were associated with higher CKD risk, while nuts, low-fat dairy products, and legumes were protective against the development of CKD) and Chen et al. [66] (who showed lower mortality in CKD patients on diet with higher plant sources).

## 5. Benefits of a Plant-Dominant Low Protein Diet

We define a plant-dominant LPD, also referred to as PLADO, as a type of LPD with DPI of 0.6–0.8 g/kg/day with at least 50% plant-based sources to meet the targeted dietary protein, and which should preferably be whole, unrefined, and unprocessed foods (Figure 3). This is consistent with the RDA of DPI of 0.8 g/kg/day, which has a high safety margin, given that based on established metabolic studies [13], the lowest DPI requirement to avoid catabolic changes is 0.45 to 0.5 g/kg/day. It has been suggested that ≥50% of DPI should be of “high biologic value” with high gastrointestinal absorbability to ensure adequate intake of essential amino acids [3]. However, other metrics, including the “protein digestibility-corrected amino-acid score,” which is a more accurate method recommended by the Food and Agricultural Organization and the World Health Organization, grant high scores to many plant-based sources and may be a more appropriate measure of protein quality [34]. Other features of PLADO include relatively low sodium intake <3 g/day, higher dietary fiber of at least 25–30 g/day, and adequate dietary energy intake (DEI) of 30–35 Cal/kg/day, assuming that the DEI calculations are based on the ideal body weight, similar to the approach to calculating DPI (Figure 3).

There are multiple pathways by which an LPD with at least 50% plant-based protein sources ameliorates CKD progression, in addition to reducing glomerular hyperfiltration [33] (Table 3):(1)Reduction in nitrogenous compounds leads to less production of ammonia and uremic toxins as an effective strategy in controlling uremia and delaying dialysis initiation [28].(2)Synergism with RAAS and SGLT2 inhibitors, since LPD reinforces the pharmaco-therapeutic effect of lowering intra-glomerular pressure through complementary mechanisms (Figure 1) [67].(3)Attenuation of metabolites derived from gut bacteria that are linked with CKD and CV disease: Animal protein ingredients including choline and carnitine are converted by gut flora into trimethylamine (TMA) and TMA N-oxide (TMAO) that are associated with atherosclerosis, renal fibrosis [68], and increased risk of CV disease and death [69]. The favorable impact on the gut microbiome [70] similarly leads to lower levels of other uremic toxins such as indoxyl sulfate and p-cresol sulfate [71].(4)Decreased acid load: plant foods have a lower acidogenicity in contrast to animal foods, and this alkalization may have additional effects beyond mere intake of natural alkali [72].(5)Reduced phosphorus burden: there is less absorbable phosphorus in plant-based proteins given the presence of indigestible phytate binding to plant-based phosphorus. Fruits and vegetables are less likely to have added phosphorus-based preservatives that are often used for meat processing [59,73,74,75].(6)Modulation of advanced glycation end products (AGE’s): higher dietary fiber intake results in a favorable modulation of AGE [76], which can slow CKD progression [77], enhance GI motility, and lower the likelihood of constipation that is a likely contributor to hyperkalemia.(7)Favorable effects on potassium metabolism: a plant-based diet based on more whole fruits and vegetables lessens the likelihood of potassium-based additives that are often found in meat products [78,79].(8)Anti-inflammatory and anti-oxidant effects: there is a decreased risk of CKD progression and CV disease due to higher intake of natural anti-inflammatory and antioxidant ingredients, including carotenoids, tocopherols, and ascorbic acid [80,81].

## 6. Features of PLADO Regimens

As stated above, the plant-dominant restricted protein diet consists of an LPD amounting to 0.6–0.8 g/kg/day with at least 50% of the dietary protein being from plant-based sources. Table 4 compares PLADO with a standard diet in the USA, in that the total amount and proportion of plant-based protein is usually 1.2–1.4 g/kg/day and 20–30%, respectively, whereas the PLADO not only has less total protein of 0.6–0.8 g/kg/day but it also includes 50% to 70% of plant-based sources for this restricted DPI goal. Hence, an 80 kg person with CKD, for instance, would be recommended to have 46 to 64 g of DPI per day, out of which 24 to 45 g will be from plant-based sources, while the rest is according to patient choice and preferences. As shown in Table 4, the total amount of animal-based protein under PLADO regimen is 14 to 32 g/day, which is less than half of the 68 to 83 g/day in the standard diet, but the patient also has the choice of being nearly or totally plant-based. There are different types of vegetarian diets [33]: (1) Vegan, or strict vegetarian (100% plant-based), diets that not only exclude meat, poultry, and seafood but also eggs and dairy products; (2) Lacto- and/or ovo-vegetarian diets that may include dairy products and/or eggs; (3) Pesco-vegetarian diets that include a vegetarian diet combined with occasional intake of some or all types of sea-foods, mostly fish; and (4) Flexitarians, which is mostly vegetarian of any of the above types with occasional inclusion of meat [33]. The PLADO does not require adherence to any of these strict diets, but is a flexible LPD of 0.6–0.8 g/kg/day range with 50% or more plant-based sources of protein based on the patient’s choice (Table 4). Whereas some nephrologists may promote a pesco-lacto-ovo-vegetarian LPD with >50% plant sources, patients have the ultimate discretion to decide about the non-plant-based portion of the protein ad lib. Based on our decades-old experience in running LPD clinics, most CKD patients will adhere to 50–70% plant-based sources, while some may choose >70% or strictly plant-based diets.

We recommend a daily sodium intake <3 g/day for a more pragmatic approach [25], as opposed to the American Heart Association’s suggested <2.3 g/day given the lack of strong evidence for the latter [25]. The PLADO regimen is CKD-patient-centric and flexible with respect to the targeted dietary goals, and is constructed based on the preferences of the patient as opposed to strict dietary regimens, with the dietitian working with patients and their care-partners to that end. Whereas we recommend a moderately low sodium intake of <3 g/day under the PLADO regimen, in those without peripheral edema and well-controlled hypertension, we have allowed slightly higher sodium intake but not greater than 4 g/day given that recent large cohort studies showed poor CKD outcomes with daily urinary sodium excretion >4 g/day [82] (Figure 3).

## 7. Safety and Adequacy of a Plant-Dominant Low-Protein Diet

Potential challenges of PLADO are outlined in Table 3, which will be largely related to the adequacy and safety of this type of dietary management of CKD patients. The risks of PEW and sarcopenia are the leading concerns, although there is little evidence for these sequelae. As discussed above and based on the U.S. recommended RDA for safe DPI ranges, it is highly unlikely that the targeted DPI of 0.6–0.8 g/kg/day with >50% plant sources will engender PEW in clinically stable individuals. No PEW was reported in 16 LPD trials cited above [13,28], including the MDRD trial [13], although PEW per se is a risk of poor CKD outcomes including faster CKD progression [83]. However, it is prudent that in patients who may develop signs of PEW or acute kidney injury (AKI), higher DPI targets should be temporarily used until PEW or AKI is resolved. On the other hand, if there is concern related to the likelihood of obesity and hyperglycemia, patients and providers should be reassured that LPD therapy in CKD has not been shown to be associated with such risks, and indeed, an LPD with plant-based sources has salutary effects on insulin resistance and glycemic index, as long as total calorie intake remains within the targeted range of 30–35 kcal/kg/day [34,55].

Another frequently stated concern is the perceived risk of hyperkalemia. We are not aware of scientific evidence to support the cultural dogma that dietary potassium restriction in CKD improves outcomes [84]. Evidence suggests that dietary potassium, particularly from whole, plant-based foods, does not correlate closely with serum potassium variability [85,86]. Indeed, a high-fiber diet enhances bowel motility and likely prevents higher potassium absorption, and alkalization with plant-based dietary sources also lowers risk of hyperkalemia [87,88,89,90,91]. Of note, dried-fruit, juices, smoothies, and sauces of fruits and vegetables require additional consideration given their high potassium concentrations. Moreover, newly available potassium-binders, which were not FDA-approved during the era of prior LPD trials such as the MDRD, may be used in the contemporary management of CKD patients at the discretion of clinicians [92].

Diet palatability and adherence to LPD or meatless diets are often cited as dietary management challenges. Based on our extensive experience in running patient-centered LPD clinics for hundreds of CKD patients [3], and given prior data on dietary adherence research [3,93], the suggested PLADO with DPI of 0.6–0.8 g/kg/day and >50% plant-based sources is feasible and well-accepted among patients with CKD [3]. Patients have the opportunity to choose the contribution of protein plant sources between 50% and 75% or >75%, and these two strata along with palatability, appetite [94], and adherence should be monitored closely in CKD clinics. If there is concern about inadequate fish intake, given data on the benefits of higher fish intake including fish oil in CKD [95,96,97], treated CKD patients can be reminded of the opportunity to consume more fish products for their remaining non-plant sources of the dietary protein. Likewise, concerns about B12 deficiency associated with meatless diets can be mitigated by the use of oral supplements as needed [98].

## 8. Impact of PLADO on Microbiome in CKD

Eating a plant-dominant, fiber-rich LPD may lead to favorable alterations in the gut microbiome, which can modulate uremic toxin generation and slow CKD progression, along with reducing cardiovascular risk in CKD patients [25,99,100,101]. Uremic plasma impairs barrier function and depletes the tight junction protein constituents of intestinal epithelium [102]. The influx of retained uremic solutes from the bloodstream per se induces changes in the microbial population simultaneous with gut wall inflammation and breakdown of epithelial junctions [103,104,105,106,107,108,109,110,111,112,113,114]. Bacterial-derived toxins then translocate back across the leaky gut barrier into the systemic circulation and promote inflammation and multi-organ dysfunction [103,115]. At least five major gut-derived uremic toxins have been associated with cardiovascular disease and mortality in CKD: indoxyl sulfate, indole-3 acetic acid, p-cresyl sulfate, TMAO, and phenylacetylglutamine [115]. In a small study that included nine CKD patients per group, which had a short duration of 6 months, LPD with or without inulin prebiotic supplementation was reported to modify the gut microbiome, increase serum bicarbonate, and improve physical function scores [116], but the investigators did not examine CKD progression or levels of gut-derived uremic toxins. Future studies should examine the role of PLADO regimens on gut microbiome in CKD patients.

## 9. Similarities and Distinctions between PLADO and other CKD Diets

In contrast to other diets used for the management of CKD, the PLADO offers a more pragmatic and patient-centered nutritional management which is aligned with contemporary dietary management goals. Unlike the diets used in the MDRD study and other studies that focused on hard outcomes, the premise of PLADO is based on its expected effects on both hard endpoints and patient-centered outcomes, including health-related quality of life, uremic symptoms, and diet palatability, while safety and adequacy remain among important goals. It is important to note that the MDRD Study was conducted in the early 1990s under dietary practices that are not relevant to contemporary practice. While high-protein diets such as keto, Atkins, and Paleo diets are popular in contemporary culture, there has been growing interest in plant-based diets across the lay and scientific communities and professional societies including the National Kidney Foundation [117], which were not considered in the MDRD Study.

Restricted protein diets that are partially to entirely plant-based are more broadly generalizable to the adult populations as compared to the prior LPD trials, including the MDRD study. PLADO can be safely recommended to both patients with early CKD, including those with any degree of proteinuria regardless of etiology [118], as well as to late-stage CKD populations, including those with an eGFR <45 mL/min/1.73 m^2^, without a lower eGFR limit, to take advantage of the effects of LPDs in controlling uremia and averting the need for dialysis. This stands in sharp contrast to the MDRD study, whose participants had relatively high eGFRs (eGFR 25 to 55 mL/min/1.73 m^2^), and which focused on slowing the progression of moderate CKD. Indeed, in the MDRD study, patients did not have diabetes [119], whereas PLADO can be non-differentially prescribed to both patients with and without diabetes with any degree of severity of CKD, consistent with the broader unmet need in the adult CKD population. It is important to note that polycystic kidney disease (PKD) patients, who usually have slower CKD progression rates, comprised 24% of the MDRD study participants [119].

## 10. Role of Dietitians in PLADO

The successful implementation of plant-based restricted protein diets is dependent on the engagement of dietitians who are well trained in the field of non-dialysis CKD [120]. Dietitians should assess regularly the dietary protein and energy as well as micronutrient intakes of CKD patients by both periodic dietary assessments and 24-h urine collections to estimate dietary intakes of macro- and micronutrients and to evaluate and improve adherence to dietary recommendations (Figure 4) [25]. Behavior change counseling by dietitians is a key skill set that is critical in successful lifestyle and habit modifications. Registered dietitians who specialize in the field of renal nutrition are usually trained to use the 24-h urine data, which may have an impact on accurate interpretation of daily nutrient intake estimates and assessment of patients adherence to the recommended medical nutrition therapy [121]. Both dietitians and other healthcare providers use telehealth increasingly frequently since the inception of COVID-19 pandemic [122]. Easy-to-use telehealth alternatives are important to overcome existing and emerging challenges in dietetic education including under the COVID-19 pandemic and other restrictions, so that patients are provided with pragmatic tools and comprehensible and consistent dietary information and skills, which fosters ownership and self-monitoring in kidney health management such as healthy kitchen approaches [122,123].

Unfortunately, however, an overwhelming majority of CKD patients never meet with a CKD-specialized dietitian prior to dialysis initiation, and most patients remain uninformed about the role of diet in disease progression and management. Among clinicians and patients, lack of awareness about the benefits of plant-dominant, low protein dietary interventions (other than low potassium diets) and available insurance reimbursement for medical nutrition therapy under guidance of a registered dietitian are significant barriers. In many regions, especially in North America and Europe, the focus and expertise of the dietitians have traditionally been centered on dialysis patient care as opposed to preventative non-dialysis dependent CKD. Past and recent reports suggest under-utilization of dietetic manpower and expertise for the purpose of non-dialysis CKD care [3]. A collective groundswell of events has recently occurred which aim to improve CKD care: the World Kidney Day focuses on reduction of the onset and progression of CKD through primary, secondary. and tertiary measures [1]; the U.S. Presidential Executive Order, “Advancing American Kidney Health” [12], refocuses kidney care from dialysis incentives to avoidance of kidney failure; the US Veterans Health Administration issued Directive 1053, “Chronic Kidney Disease: Prevention, Early Recognition, and Management,” establishes federal policy targeting CKD prevention through integrated care including medical nutrition therapy [124], and the advocacy of renal dietitians for patient-centric LPD regimens containing fewer animal products and more plant-based sources of protein such as PLADO [125]. This is a sharp contrast to prior LPD recommendations with less flexible regimens such as strict plant-based dieting or very low DPI of <0.4 g/kg/day combined with supplements, that may be less palatable, unsustainable, and non-pragmatic for broad application in the real-world scenarios of CKD patient care.

## 11. Recommendations for Practical Implementation of PLADO

After the first 3 months, which includes preliminary education on LPD regimens with >50% plant sources and acquiring food preparation skills, participating CKD patients should be assessed every 3 to 6 months by the dietitian. During each visit, dietary re-education along with dietary assessment should be conducted and patient’s progress in reaching the goals should be examined. In line with the pragmatic nature of PLADO regimens, the dietary re-education and follow-up visits can be performed in parallel with routine follow-up CKD clinic visits on the same days of the ambulatory clinic appointments, thus avoiding the burden of additional diet-related travels to the CKD clinic. In addition to in-person visits, there could be monthly to tri-monthly telephone calls with the CKD patients under CKD therapy, or even more frequently if needed, to reinforce diet planning and adherence and to answer questions about preparation of plant-dominant meals and cooking questions. Adherence to PLADO should be evaluated by comparing the LPD goals, i.e., 0.6–0.8 g/kg/day and >50% plant sources, to the estimated DPI using 24-hr urine nitrogen (see below) and 3-day diet assessments, respectively. Complementary dietary education of the patients and their care-partners should be provided both during the face-to-face visits and via phone calls.

The specialized knowledge and services of a renal dietitian ensure accurate nutrition education, meal planning and evaluation of body composition to sustain health. Components of a CKD nutrition evaluation may include the following (see Table 5): (1) Dietary education for LPD with >50% plant-based protein sources, (2) Dietary assessment using a three-day diet diary with interview, (3) Simplified anthropometry that includes triceps and biceps skinfolds [126] and mid-arm circumference [127], (4) Body fat estimation using either bioimpedance analyses or near-infra red interactance [128,129,130], (5) The Malnutrition-Inflammation Score (MIS) [131,132,133,134], including Subjective Global Assessment [135], and (6) Handgrip strength test [136]. The dietary education along with the above evaluations usually take 30 min to one hour of the dietitian’s time during each visit according to our previous and ongoing nutritional clinic operations.

## 12. Concurrent Pharmacotherapy and Other Interventions

Regardless of the type of the dietary regimen, participation in the PLADO plan does not interfere with any other aspects of the CKD patient care including prescribed medications such as angiotensin pathway modulators, other anti-hypertensive medications, anti-diabetic medications such as SGLT2 inhibitors, phosphorus binders, potassium binders, sodium bicarbonate, etc. Indeed, it is expected that dietary protein restriction will have a synergistic effect on these pharmacotherapies [67]. The inclusion of plant-based foods should not necessitate a reduction in any of these medications over time.

## 13. Laboratory Tests for Nutritional Management of CKD

Consistent with the pragmatic and cost-efficient nature of the PLADO regimen, all relevant laboratory tests are performed in the clinical laboratories of the respective medical centers typically as part of routine CKD care. With the exception of a semi-annual serum vitamin B12 level, quarterly to semi-annual laboratory tests include routine chemistry panels (including serum Na, K, CO_2_, Cl, urea, creatinine, glucose), liver function tests, hemoglobin A1c, anemia and mineral and bone disorders (MBD) parameters including calcium, phosphorus, and parathyroid hormone. Urinalysis and spot urine for urinary protein/albumin and creatinine should be tested, and eGFR is calculated [137]. Participating patients are instructed to collect 24-h urine samples according to the directions that should be repeated during each ambulatory visit and/or each phone call, i.e., not collecting the first AM urine of Day 1, collecting the first AM urine of Day 2 as the last collection component, and the entire micturition in-between. The 24-hr urine should include measurements of urine urea nitrogen (UUN), sodium (UNa), potassium UK, creatinine (UCr), albumin, and protein, as well as urine volume (UV). The following measures should be calculated and reviewed by both the nephrologist and dietitian during each visit [25]: (1)Creatinine clearance: **UCr*UV/SCr** in ml/min, and to compare to eGFR;(2)Creatinine index: **UCr/Weight (mg/kg)**, to identify 24-h urine collection inaccuracies including under- and over-collections by comparing it to the expected value of 1–2 mg/kg/d for women and 1.5–2.5 mg/kg/day for men;(3)Estimated DPI (eDPI): **UUN*6.25+0.03*weight** (in g/kg/day); for patients with substantial proteinuria >3 g/day, the daily proteinuria amount is added to the above eDPI [3,25];(4)Estimated dietary Na intake: **UNa in mmol/44** (g/day);(5)Estimated dietary K intake: **UK in mmol/25** (g/day);(6)24-h urinary protein and albumin excretion (mg/day).

See Table 5 for the overview of the laboratory tests.

## 14. Suggested Self-Administered Questionnaires

Based on the goals and the extent of the operation and resources of the CKD clinic, some to all of the following self-administered questionnaires can be used during each or alternating ambulatory visit: (1) Diet, Palatability, and Appetite Questionnaire: the appetite component allows grading appetite and recent changes [138]. The palatability component includes 12 items and grades palatability and feasibility of dietary intervention [138]. These items are combined with diet assessment of the HEMO Study [139]. (2) Quality of life KDQOL™ including SF36: This has been used and validated extensively [133]. (3) Uremic symptoms questionnaire: This questionnaire is derived from the “Symptom Assessment Instrument” by Weisbord et al. [140], which was created and validated in US veterans with stage 5 CKD. (4) Self-Perception and Relationship Questionnaire: This item will assess the psychosocial-spiritual well-being using the 28-item scale [141]. (5) Food Frequency Questionnaire (FFQ) [142]: this questionnaire has been developed by Kalantar-Zadeh et al. using the Block FFQ from UC Berkeley, and can be used semi-annually to annually (see Table 5).

## 15. Diet Safety and Transient Dietary Regimen Suspension

Once a patient has completed the 3-month run-in period including dietary education and food preparation training and adjustments, there should be periodic (every 3–6 months) ambulatory visits with continued data collection and review. If PEW signs are observed, or in case of an event that requires suspension of the LPD such as hospitalization with AKI, regardless of dialysis need, or major adverse cardiovascular events (MACE), the LPD can be transiently suspended, and the patient can resume the LPD and the study protocol at a later time, usually within 90 days of the suspension of the dietary regimen participation if deemed safe. Serum potassium levels >5.5 mEq/L will preferentially be managed by potassium-binders (first line) and/or reducing the potassium-rich components of the diet (second line), as opposed to the current standard of care in that traditional low-potassium dietary adjustments are pursued as the main approach, followed as needed by the administration of potassium-binders.

## 16. Challenges and Pitfalls of the Dietary Management of CKD

As stated above, the proposed plant-dominant diet may cause hyperkalemia and could thus be hazardous to patients with advanced CKD. Nephrologists and dietitians should closely monitor patients during the 3-month run-in period and thereafter for adverse events. Dietitian support is necessary for appropriate education on culinary strategies to reduce excessive potassium content while preserving flavor and nutrition. Physicians should take appropriate actions including the use of potassium-binders or suspension of the patient’s participation if this should be the safest approach. We do not expect that most patients on plant-based diets will develop hyperkalemia, as these diets are alkalinizing and alter intestinal transit time (see above), especially if dried fruit, juices, smoothies, and sauces can be minimized or avoided along with judicious avoidance of processed food with added potassium-based additives and preservatives [92]. Those who are extremely prone to develop hyperkalemia would display this early in the course of the intervention and the PLADO would be discontinued if hyperkalemia cannot be controlled. Less constipation as a result of PLADO is associated with favorable cardiovascular and renal outcomes [143,144].

It is important to note that the emerging standard of care in CKD is a restricted protein diet of 0.55–0.6 g/kg/day for non-diabetic CKD and 0.6–0.8 g/kg/day for diabetic CKD according to the updated KDOQI nutrition guidelines as of September 2020 [145], and if this is implemented, this is in support of our PLADO regimen. Whereas it is true that an LPD should be the stated goal according to the 2020 KDOQI guidelines, this is typically not followed in everyday clinical practice, where dietary interventions are driven by biochemical abnormalities such as hyperkalemia or hyperphosphatemia. Indeed, prior KDOQI guidelines had recommended DPI of 0.8 g/kg/day without any clear range, which is rarely pursued in a real-world scenario.

It could be argued that under the PLADO regimen there is no clear meal plan. However, the reasonably wide dietary protein range of 0.6–0.8 g/kg/day along with the recommended plant-based proportion of 50% or higher ensures the intended flexibility and pragmatism of the PLADO regimen, so that further adjustments to individualized characteristics of different patients can be implemented according to the principles of the “precision nutrition” as also shown for the dietary management of diabetes [146]. Some health care providers, as well as patients, may express concerns that the carbohydrates burden of plant-dominant diets confounds dietary management of obesity, metabolic syndrome, or diabetes. However, different types of carbohydrates have different glycemic indices, and high protein or ketogenic diets, which may be recommended for these conditions, are associated with untoward consequences in disease and health [21]. Complex carbohydrates, including whole and minimally processed foods, are high in fiber and antioxidants and can reduce insulin resistance and improve glycemic control by a variety of biologically plausible mechanisms [147]. Indeed, whole food plant-based diets can help reduce weight in overweight and obese persons and help improve the lipid profile and other risk factors related to cardiovascular disease or diabetes [148], and they are also more cost-effective than meat-dominant foods [149]. Indeed, plant-based foods are less expensive than animal based foods, including meat and poultry, in terms of cost per serving [150]. The patient and the dietitian should work together in establishing a patient-specific “Healthy Kitchen for CKD” and patients and their care-partner should gain experience in implementing patient-centered dietary interventions for CKD management. Careful and balanced industry partnership can be sought to develop innovative “Healthy Kidney Diet Plans” to help people with CKD change their diet to delay the progression of the disease and to defer and prevent kidney failure.

It has been argued that many people with CKD enjoy eating high amounts of meat, and it is highly unlikely that they will adopt an LPD with >50% plant-sources, especially since many dietitians recommend a high protein diet as an approach against obesity and diabetes. Several authors of this review paper, including both nephrologists and dietitians, have successfully implemented an LPD and plant-dominant diet education in CKD in their respective medical centers. They are aware of the cultural and dietary challenges, including in Americans and other Westerners, as described in their published reports [3], and have been able to introduce and implement the PLADO regimens as described here.

Another potential challenge is the misconception related to the definition of the conservative management of advanced CKD, which is often confused with palliative and supportive care towards the end of life and without requiring special diets. This incorrect assumption is the result of confusing different types of conservative management of CKD, and their similarities and distinctions that have recently been better clarified [9], in that a dietary approach including PLADO is a “preservative” management of CKD and a life-sustaining and kidney rejuvenating alternative.

## 17. Anticipated Impact and Future Steps

Our proposed PLADO regimen, which has been successfully implemented in several centers in the USA, reinvigorates the role of diet and nutrition in CKD management and may have major clinical and public health implications among numerous populations who are at risk for or have underlying CKD, as well as millions of Americans and people around the world with these conditions. The discussions about plant-dominant diets such as PLADO will also lead to a generation of critical data about the efficacy and safety of plant-dominant regimens, and will challenge the prevailing dialysis-centered paradigm. It is also aligned with recent US national directives, such as the 2020 VA CKD Directive, promoting medical nutrition therapy, and the July 2019 Executive Order’s restructuring of the ESKD program by preemptively involving patients and dietitians in earlier phases of CKD care, rather than dialysis preparation. This model stands in sharp contrast to the current payment system whereby the renal dietitians’ focus of work is in the dialysis units, while patients at risk of kidney failure have little or no access to nutritional support. The PLADO regimen also innovatively emphasizes the important skillset provided by trained dietitians and other healthcare providers in CKD patient care outside the dialysis arena. Averting and delaying dialysis will also result in major cost benefits to health care systems and likely improve patient longevity and health-related quality of life.

Whereas well-designed, pragmatic randomized controlled trials are warranted to verify the efficacy of PLADO in achieving improvement in clinical end points, this dietary regimen can be used safely for the management of CKD. PLADO has the advantage of considering both dietary protein quantity (LPD) and quality (>50% plant-based), instead of quantity alone or being solely plant-based. Its unique pragmatic design efficiently leverages CKD clinic visits and hands-on involvement of nephrologists and dietitians during routine ambulatory nephrology assessments, providing unique feasibility to conduct CKD management successfully. Finally, examining mastery of self-management skills through “Teach-to-Goal” under the “Healthy Kidney through your Kitchen” program by dietitians enable patients with CKD to more effectively self-manage their diet and kidney disease [3].

## Figures and Tables

**Figure 1 nutrients-12-01931-f001:**
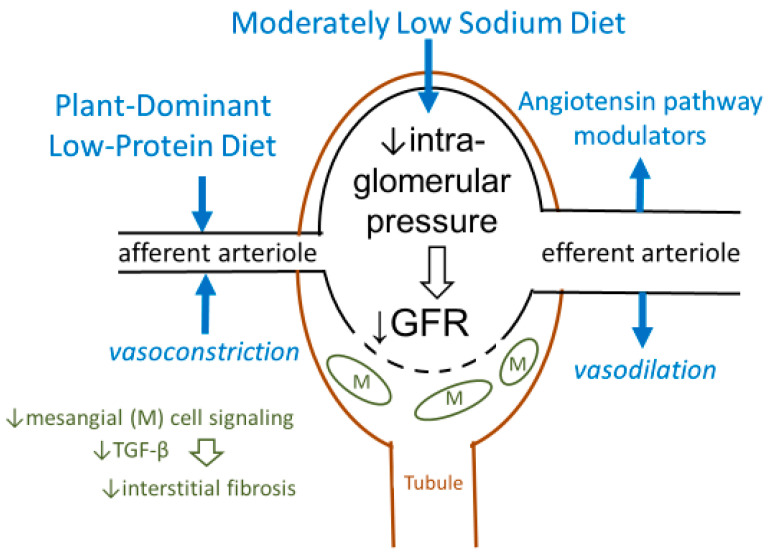
Effects of a plant-dominant low-protein diet on afferent arteriole contraction leading to reduced intra-glomerular pressure and nephron longevity (adapted from Kalantar-Zadeh and Fouque, *N Engl J Med* 2017) [25].^.^

**Figure 2 nutrients-12-01931-f002:**
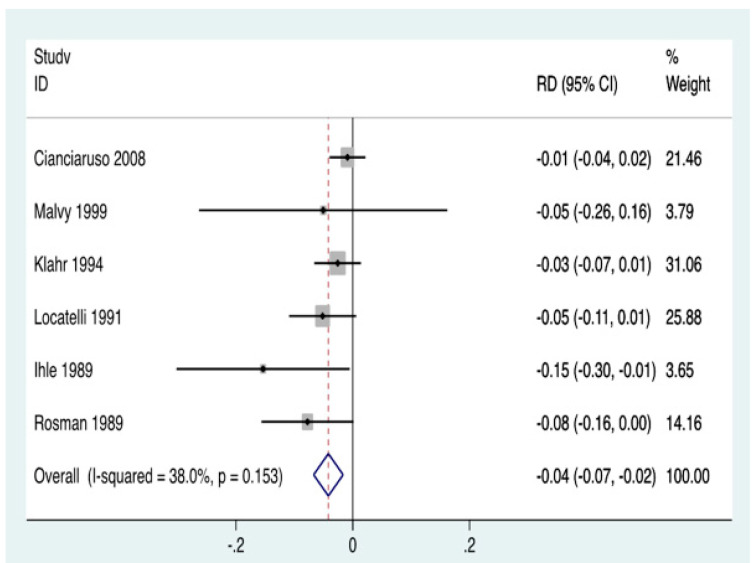
Meta-analysis of the randomized controlled trials with low protein diet suggesting efficacy of diet in lowering the risk of kidney failure. This meta-analysis includes six (out of 16) randomized control trials of low protein diet (adapted from Rhee et al., *J Cachexia Sarcopenia Muscle* 2018) [28].

**Figure 3 nutrients-12-01931-f003:**
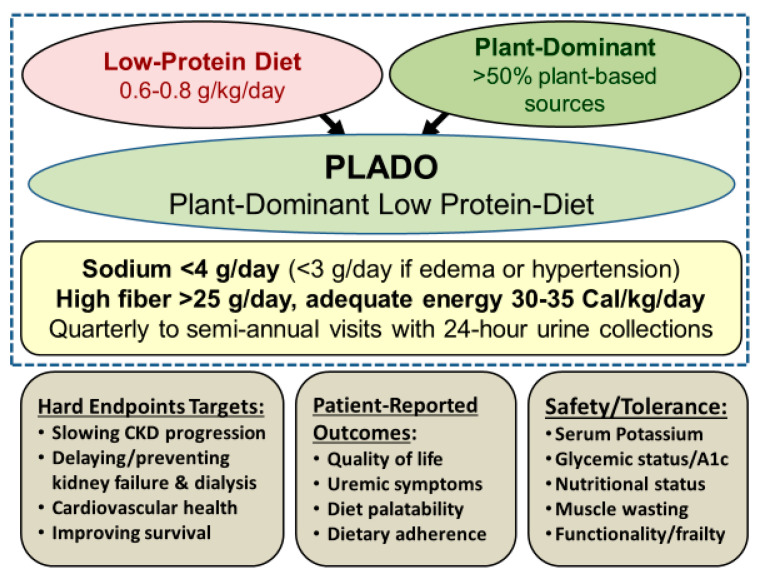
Overview of the plant-dominant low-protein diet (PLADO) for nutritional management of CKD, based on a total dietary intake of 0.6–0.8 g/kg/day with >50% plant-based sources, preferentially unprocessed foods, relatively low dietary sodium intake <3 g/day (but the patient can target to avoid >4 g/day if no edema occurs with well controlled hypertension), higher dietary fiber of at least 25–30 g/day, and adequate dietary energy intake of 30–35 Cal/kg/day. Weight is based on the ideal body weight. Note that serum B12 should be monitored after three years of vegan dieting.

**Figure 4 nutrients-12-01931-f004:**
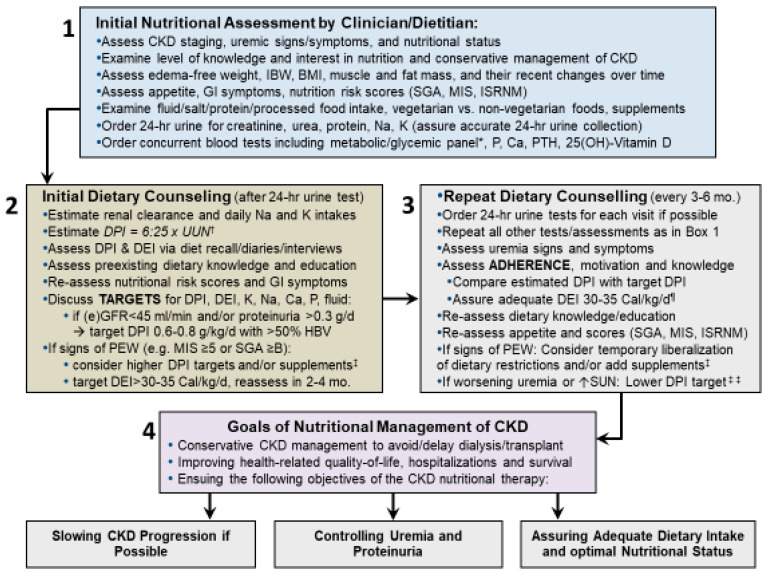
An algorithm and steps for the approach to the nutritional management of patients with CKD. Note that in addition to direct dietary assessments, periodic 24-h urine collections should be used to estimate dietary protein, sodium, and potassium intakes in order to assess adherence to dietary recommendations (adapted from the Supplementary-Appendix-Figure S4. Under Kalantar-Zadeh and Fouque, *N Engl J Med.* 2017) [25]. * Comprehensive metabolic and glycemic panels include electrolytes, SUN, creatinine, glucose, hemoglobin A1c, liver function tests, and the lipid panel. ^†^ The full equation is: DPI = 6:25 × UUN + 0:03 × IBW ^†^ Add the amount of daily proteinuria in grams if proteinuria >5 g/d. Calculate the creatinine index (24-hr urine creatinine divided by actual weight or IBW if obese) and compare it to the expected value of 1–1.5 g/kg/d for women and 1.5–2 g/kg/day for men. ^‡^ Dietary supplements can be added to provide additional sources of energy and/or protein including—but not limited to—CKD specific supplements, essential amino-acids, or keto-analogues (ketoacids) of amino-acids. ^¶^ To ensure adequate DEI of at least 30–35 Cal/kg/d, higher fat intake can be considered, e.g., non-saturated fats, omega 3-rich flaxseed, canola, and olive oil. ^‡‡^ If worsening uremic signs and symptoms occur, DPI < 0.6 g/kg/d with supplements can be considered. Abbreviations: BMI: body mass index, CKD: chronic kidney disease, d: day, DEI: dietary energy intake, DPI: dietary protein intake; eGFR: estimated glomerular filtration rate, GI: gastrointestinal, HBV: high biologic value, IBW: ideal body weight, ISRNM: International Society of Renal Nutrition and Metabolism, K: potassium; MIS: malnutrition–inflammation score; Na: sodium; Phos.: phosphorus; PTH: parathyroid hormone, PEW: protein energy wasting, SGA: subjective global assessment, SUN: serum urea nitrogen, UUN: urine urea nitrogen.

**Table 1 nutrients-12-01931-t001:** Selected studies of high-protein and kidney function. DPI: dietary protein intake; CKD: chronic kidney disease.

Study (Year)	Cohort, [N] (Country)	Duration Of Follow Up	Findings
Esmeijer [22] (2020)	Alpha Omega Cohort (2255) (Netherlands)	41 mo	↑ DPI 0.1 g/kg/day associated with ↑ eGFR decline of −0.12 ml/min/year
Jhee [23] (2020)	South Korea (9226)	14 yrs	3.5-fold ↑ risk of hyperfiltration. 1.3-fold ↑ faster decline
Malhotra [24] (2018)	Jackson Heart (USA) (5301)	8 yrs	↑ DPI density associated with ↑ eGFR decline
Farhadnejad [24] (2018)	Healthy Iranian adults (1797)	6.1 yrs	48% ↑ risk of incident CKD in high DPI

**Table 2 nutrients-12-01931-t002:** Low protein diet (LPD)-controlled trials with greater than 30 participants in each study [25].^.^

Study (Year)	Participants	Diet (g/kg/day)	Duration of Follow Up	Results
Rosman (1984) [35,36].	247 CKD 3–5 pts	0.90–0.95 vs. 0.70–0.80 vs. unrestricted	4 yrs	Significant CKD slowing in LPD in male pts.
Ihle (1989) [37]	72 CKD 4–5 pts	LPD (0.6) vs. higher DPI (0.8)	18 mo	Loss of GFR in control vs. LPD (*p* < 0.05). Wt loss
Lindenau (1990) [38]	40 CKD 5 pts	LPD vs. sVLPD (0.4) w KA	12 mo	Decreased phos. with sVLPD and improved bone health
Williams (1991) [39]	95 CKD 4–5	LPD (0.7) vs. 1.02–1.14	18 mo	No differences, minor Wt loss
Locatelli (1991) [40]	456 CKD 3–4	0.78 vs. 0.9	2 yrs	Trend for difference in renal outcomes (*p* = 0.059).
MDRD Klahr (1994) [41]	585 CKD 3–4	1.3 vs. 0.6	27 mo	No difference in GFR decline at 3 years.
Montes-Delgado (1998) [42]	33 CKD 3–5	LPD vs. sLPD	6 mo	Slower eGFR decline with supplements
Malvy (1999) [43]	50 CKD 4–5	sVLPD (0.3) KA vs. LPD (0.65)	3 yrs	Decreased SUN lean body mass and fat in sVLPD
Teplan (2001) [44]	105 CKD 3b–4	LPD w vs. w/o KA	3 yrs	Slower CKD progression
Prakash (2004) [45]	34 CKD 3b–4	0.6 vs. 0.3 w KA	9 mo	Faster decline in LPD
Brunori (2007) [46]	56 > 70 yrs old CKD 5	sVLPD (0.30) w KA vs. dialysis	27 mo	Similar survival but more hospitalizations in dialysis
Mircescu (2007) [47]	53 CKD 4–5	sVLPD (0.3) vegan w KA vs. LPD	48 wks	Less dialysis initiation in sVLDP
Cianciaruso (2008) [48]	423 CKD 4–5	0.55 vs. 0.80	18 mo	Reduced urinary urea, Na, phos
Di Iorio (2009) [49]	32 CKD w proteinuria	VLPD vs. LPD	6 mo	58% greater reduction in proteinuria
Jiang (2009 and 2011) [50,51].	60 PD w RKF	LPD vs. sLPD w KA vs. HPD	12 mo	RKF decreased in the LPD and HPD.
Garneata (2016) [52]	207 CKD 4–5	LPD (0.6) vs. sVLPD w KA	15 mo	Less dialysis initiation

Abbreviations: Pts.: patients, yrs: years, mo: months, Et: weight, phos.: phosphorus, sVLPD: supplemented very low protein diet.

**Table 3 nutrients-12-01931-t003:** Benefits and challenges of LPD with >50% plant-based protein sources.

Benefits of LPD with >50% Plant Sources	Potential Challenges of LPD
Lowering intra-glomerular pressure	Risk of protein-energy wasting (PEW)
Synergistic effect with RAASi and SGLT2i	Inadequate essential amino acids
Controlling uremia and delaying dialysis	Undermining obesity management
Preventing cardiovascular harms of meat	High glycemic index
Less absorbable phosphorus	High potassium load and hyperkalemia
Lowering acid-load with less acidogenicity	Low palatability and adherence
High dietary fiber enhancing GI motility	Inadequate fish intake if vegan
Favorable changes in microbiome	
Less TMA N-oxide (TMAO), leading to less kidney fibrosis	
Less inflammation and oxidative stress	

**Table 4 nutrients-12-01931-t004:** Comparing Low Protein Diet (LPD) >50% plant-based protein sources. Known as PLADO, versus standard diet, based on 2400 Cal/day in an 80-kg person.

Protein Metric	Standard Diet	LPD >50% Plant-based Sources (PLADO)
Proportion of plant-based protein, %	20–30%	50–70% *
Total protein per kg IBW, g/kg/day	>0.8, usually 1.2–1.4	0.6–0.8
Total protein intake, g/day	96 to 112 g	48 to 64 g
Protein density, g/100 Cal	4.4–5.1	2.2–2.9
Proportion of energy from protein, %	16–19%	8–11%
Total plant-based protein, g/day	24–34	24–45
Total animal-based protein, g/day	68–83	14–32 (or none *)

* up to 100% vegan is allowed based on patient choice.

**Table 5 nutrients-12-01931-t005:** Overview of the recommended ambulatory visits and tests under the PLADO regimen (* these items are more relevant to sophisticated centers or under research protocols).

	Timeline of for PLADO Therapy Visits	“Run-in” Period			Year 1 (Quarterly)			Years 2+ (Semi-Annual)			Needed Time	
	PALDO Months	0	1	3	6	9	12	18	24	30	36	
	History and physical examination with updates on clinical and dietary status	X	x	x	x	x	X	x	x	X	x	10–20 min
Lab tests	Routine lab panel: CMP/LFT, anemia, MBD, A1c	X		x	x	x	X	x	x	X	x	<10 min
	Spot urine, urinalysis, protein, albumin, creatinine	X		x	x	x	X	x	x	X	x	<5 min
	24 hr urine: Nitrogen, Na, K, creatinine, alb, prot.	X		x	x	x	X	x	x	X	x	Collected at home
	eGFR assessment and creatinine and urea clearance	X		x	x	x	X	x	x	X	x	
Dietitian visit	Dietary education for LPD >50% plant based	X	x	x	x	x	X	x	x	X	X	10–20 min
	Dietary assessment, three-day diet diary with interview	X	x	x	x	x	X	x	x	X	X	10–20 min
	Anthropometry: triceps and biceps skinfolds, mid-arm circumference *	X		x	x	x	X	x	x	X	X	2–4 min
	Body fat estimation *	X		x	x	x	X	x	x	X	X	1–2 min
	Malnutrition-inflammation score *	X		x	x	x	X	x	x	X	x	2–5 min
	Handgrip strength test *	X		x	x	x	X	x	x	X	x	1–2 min
	Phone calls to reinforce PLADO education, adherence, and meal preparation	x	x	x	x	x	X	x	x	X	x	10–30 min
Questionnaires	Diet palatability and appetite questionnaire	x	x	x	X	x	X	x	x	X	x	15–30 min
	Food Frequency Questionnaire *	x			x		X	x	x	X	x	15–30 min
	Quality of life: KDQOL™ including SF36 quest *	x		x	x	x	X	x	x	X	x	10–15 min
	Uremic symptoms questionnaire	x		x	x	x	X	x	x	X	x	10–15 min
	Self-Perception and Relationship Questionnaire *	x		x	x	x	X	x	x	X	x	10–15 min

Abbreviations: RD: Registered dietitian, CMP: comprehensive metabolic panel, LFT: liver function tests, MBD: Mineral and bone disease markers, Na: sodium, K: potassium, Na, K, creatinine, alb: albumin, prot.: protein, KDQOL: Kidney disease quality of life, SF36: Short Form with 36 items of quality of life.

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
