# Peer review of "Plant-Dominant Low-Protein Diet for Conservative Management of Chronic Kidney Disease"

_nutrients, 2020, doi:10.3390/nu12071931_

Round 1
Reviewer 1 Report
The manuscript by Kalantar-Zadeh and others is a comprehensive review of the literature to support the use of Plant-Dominant Low-Protein Diet for treatment of chronic kidney disease. Overall, the piece is of high-quality and fascinating to read. In addition to describing the benefit of low-protein diets in treating CKD, the authors have also put significant efforts in constructing a detailed algorithm for effectively implementing PLADO in clinics. The manuscript should be qualified for publication pending a minor revision.
It would be helpful to have some discussions regarding the effects of low-protein diets particularly PLADO on short- and long-term blood glucose control and insulin sensitivity. This is important to clear doubts in many clinicians and/or dieticians who prefer high-protein, low carb diets to manage obesity/diabetes and related complications including diabetic nephropathy.
Line 440: "non-diabetic CD" should be "non-diabetic CKD"
Author Response
Reviewer 1 - Comments and Suggestions for Authors
- The manuscript by Kalantar-Zadeh and others is a comprehensive review of the literature to support the use of Plant-Dominant Low-Protein Diet for treatment of chronic kidney disease. Overall, the piece is of high-quality and fascinating to read. In addition to describing the benefit of low-protein diets in treating CKD, the authors have also put significant efforts in constructing a detailed algorithm for effectively implementing PLADO in clinics. The manuscript should be qualified for publication pending a minor revision.
OUR RESPONSE: We appreciate very much the evaluation and suggestions by Reviewer 1.
- It would be helpful to have some discussions regarding the effects of low-protein diets particularly PLADO on short- and long-term blood glucose control and insulin sensitivity. This is important to clear doubts in many clinicians and/or dieticians who prefer high-protein, low carb diets to manage obesity/diabetes and related complications including diabetic nephropathy.
OUR RESPONSE: We appreciate this important suggestion. We understand that some health care providers, including physicians and dietitians, as well as patients may think that all carbohydrates are unfavorable and should be replaced by high protein diet, and it is said that because a plant-based diet is carbohydrate dominant, it should not be recommended. There are indeed different types of carbohydrates including complex carbohydrates, which are whole and minimally processed foods, that are to be recommended to persons with diabetes, obesity or metabolic syndrome. As a relevant example, the same plant-based item may be prepared and presented as dietarily favorable (like a fresh banana) or unfavorable (like banana bread). Hence it is important to differentiate different types and aspects of plant-based diet given the vast diversity, similar to different dietary fats that also include more or less favorable items. One of our co-authors (Dr Shivam Joshi) recently wrote a short viewpoint in JAMA Int Med about this false notion as it relates to ketogenic diets.1 Moreover, there are other studies as the one by McMacken et al2 that suggests diets based in whole and minimally processed plant foods reduce insulin resistance and improve glycemic control by a variety of proposed mechanisms. Plant-based diets are high in fiber, antioxidants, and magnesium, all of which have been shown to promote insulin sensitivity. Indeed, whole food plant-based (WFPB) diet can lead to significant reduction in weight and serum cholesterol as well as other risk factors related to cardiovascular disease or diabetes.3 Hence, we have revised the following paragraph in the amended manuscript, starting from Line 453:
It could be argued that under the PLADO regimen there is no clear meal plan. However, the reasonably wide dietary protein range of 0.6-0.8 g/kg/day along with the recommended plant-based proportion of 50% or higher ensures the intended flexibility and pragmatism of the PLADO regimen, so that further adjustments to individualized characteristics of different patients can be implemented according to the principles of the “precision nutrition” as also shown for the dietary management of diabetes.4 Some health care providers, as well as patients, may express concerns that the carbohydrates burden of plant-dominant diets confounds dietary management of obesity, metabolic syndrome, or diabetes. However, different types of carbohydrates have different glycemic index, and high protein or ketogenic diets, that may be recommended for these conditions, are associated with untoward consequences in disease and health.1 Complex carbohydrates, including whole and minimally processed foods, are high in fiber and antioxidants and can reduce insulin resistance and improve glycemic control by a variety of biologically plausible mechanisms.2 Indeed, whole food plant-based diet can help reduce weight in overweight and obese persons and help improve lipid profile and other risk factors related to cardiovascular disease or diabetes,3 and they are also more cost-effective than meat-dominant foods.5 The patient and the dietitian should work together in establishing a patient specific “Healthy Kitchen for CKD” and patients and their care-partner should gain experience in implementing patient-centered dietary interventions for CKD management. Careful and balanced industry partnership can be sought to develop innovative “Healthy Kidney Diet Plans” to help people with CKD change their diet to delay the progression of the disease and to defer and prevent kidney failure.
- Line 440: "non-diabetic CD" should be "non-diabetic CKD"
OUR RESPONSE: We appreciate for identifying this error, which is now corrected.
Once again, we appreciate very much the evaluation and suggestions by Reviewer 1.
References for this response letter:
- Joshi S, Ostfeld RJ and McMacken M. The Ketogenic Diet for Obesity and Diabetes-Enthusiasm Outpaces Evidence. JAMA internal medicine. 2019;179(9):1163-1164. doi: 10.1001/jamainternmed.2019.2633. PubMed PMID: 31305866. URL: https://www.ncbi.nlm.nih.gov/pubmed/31305866.
- McMacken M and Shah S. A plant-based diet for the prevention and treatment of type 2 diabetes. J Geriatr Cardiol. 2017;14(5):342-354. doi: 10.11909/j.issn.1671-5411.2017.05.009. PubMed PMID: 28630614; PMCID: PMC5466941. URL: https://www.ncbi.nlm.nih.gov/pubmed/28630614.
- Wright N, Wilson L, Smith M, Duncan B and McHugh P. The BROAD study: A randomised controlled trial using a whole food plant-based diet in the community for obesity, ischaemic heart disease or diabetes. Nutr Diabetes. 2017;7(3):e256. doi: 10.1038/nutd.2017.3. PubMed PMID: 28319109; PMCID: PMC5380896. URL: https://www.ncbi.nlm.nih.gov/pubmed/28319109.
- Wang DD and Hu FB. Precision nutrition for prevention and management of type 2 diabetes. The lancet Diabetes & endocrinology. 2018;6(5):416-426. doi: 10.1016/S2213-8587(18)30037-8. PubMed PMID: 29433995. URL: https://www.ncbi.nlm.nih.gov/pubmed/29433995.
- Tuso PJ, Ismail MH, Ha BP and Bartolotto C. Nutritional update for physicians: plant-based diets. Perm J. 2013;17(2):61-6. doi: 10.7812/TPP/12-085. PubMed PMID: 23704846; PMCID: PMC3662288. URL: https://www.ncbi.nlm.nih.gov/pubmed/23704846.
- Sugimoto M, Asakura K, Masayasu S and Sasaki S. Relationship of nutrition knowledge and self-reported dietary behaviors with urinary excretion of sodium and potassium: comparison between dietitians and nondietitians. Nutr Res. 2016;36(5):440-51. doi: 10.1016/j.nutres.2015.12.012. PubMed PMID: 27101762. URL: https://www.ncbi.nlm.nih.gov/pubmed/27101762.
- Kalantar-Zadeh K and Moore LW. Impact of Nutrition and Diet on COVID-19 Infection and Implications for Kidney Health and Kidney Disease Management. J Ren Nutr. 2020;30(3):179-181. doi: 10.1053/j.jrn.2020.03.006. PubMed PMID: 32291198; PMCID: PMC7186539. URL: https://www.ncbi.nlm.nih.gov/pubmed/32291198.
- Fulgoni V, 3rd and Drewnowski A. An Economic Gap Between the Recommended Healthy Food Patterns and Existing Diets of Minority Groups in the US National Health and Nutrition Examination Survey 2013-14. Front Nutr. 2019;6:37. doi: 10.3389/fnut.2019.00037. PubMed PMID: 31019912; PMCID: PMC6458255. URL: https://www.ncbi.nlm.nih.gov/pubmed/31019912.
Reviewer 2 Report
This work provides useful information about the plant-based dietary intake in CKD patients. The paper explains all aspects of PLADO regiment which can be useful for both nephrologists and dietitians.
Please see below the three major comments that I have:
-Although authors have explained about concerns with hyperkalemia, the explanation is not consistent throughout the manuscript. For instance, in line 239, authors explain about lack of evidence about potassium intake restriction and CKD outcomes. However, as authors have mentioned as well, the concern with increased serum potassium always remains when it comes to higher intakes of certain fruits and vegetables. Please make sure that you would have a complete and consistent message about intake of fruits and vegetables, high in potassium, in CKD patients and how they should be monitored for risk of hyperkalemia.
-Line 295: Authors explain about role of dietitians in PLADO; although the suggested steps are very useful and actually needed, in some cases it can be unrealistic. For example use of telehealth for this purpose, or periodic dietary assessment and 24 hour urine collections. Please re-word this section to be more reflective of how much dietitians can be really involved in this process.
-Although authors have explained about the implementation of this diet, it is still unclear if it is possible for a CKD patient to really use this diet in long-term. For example, in most cases, budget is one of the important factors which dominants the dietary intake. In general, plant-based diets can be more expensive than a typical Western diet. Please explain how the socioeconomic status of CKD patients (and their families) will be taken into consideration when it comes to PLADO regiment.
Author Response
Reviewer 2 - Comments and Suggestions for Authors
- This work provides useful information about the plant-based dietary intake in CKD patients. The paper explains all aspects of PLADO regiment which can be useful for both nephrologists and dietitians.
OUR RESPONSE: We appreciate very much the evaluation and suggestions by Reviewer 2.
- Please see below the three major comments that I have:
-Although authors have explained about concerns with hyperkalemia, the explanation is not consistent throughout the manuscript. For instance, in line 239, authors explain about lack of evidence about potassium intake restriction and CKD outcomes. However, as authors have mentioned as well, the concern with increased serum potassium always remains when it comes to higher intakes of certain fruits and vegetables. Please make sure that you would have a complete and consistent message about intake of fruits and vegetables, high in potassium, in CKD patients and how they should be monitored for risk of hyperkalemia.
OUR RESPONSE: We appreciate Reviewer 2’s important comments about the challenge of potassium burden and the need for a more coherent approach. We believe that the notion that plant-based foods, especially fresh fruits and vegetables, contribute to higher potassium burden in CKD is rapidly changing, and there have been increasingly larger numbers of recent published studies in this regard. Plant based diets are shown to have minimal risk of hyperkalemia if well planned, and especially if renal dietitians provide the needed support with more focus on added sources of potassium such as avoiding K-based preservatives, among others. A single diet does not meet the needs of all person with CKD, and individualizing the diet based on specific needs and characteristics of these patients in important, as also being increasingly more emphasized under the principle of the “Precision Medicine”. Hence, we have revised the following paragraph in the amended manuscript, starting from Line 453:
It could be argued that under the PLADO regimen there is no clear meal plan. However, the reasonably wide dietary protein range of 0.6-0.8 g/kg/day along with the recommended plant-based proportion of 50% or higher ensures the intended flexibility and pragmatism of the PLADO regimen, so that further adjustments to individualized characteristics of different patients can be implemented according to the principles of the “precision nutrition” as also shown for the dietary management of diabetes.4
- -Line 295: Authors explain about role of dietitians in PLADO; although the suggested steps are very useful and actually needed, in some cases it can be unrealistic. For example use of telehealth for this purpose, or periodic dietary assessment and 24 hour urine collections. Please re-word this section to be more reflective of how much dietitians can be really involved in this process.
OUR RESPONSE: We appreciate Reviewer 2 for this important point about the role of dietitians. Several (N=5) coauthors of this article are practicing Registered Dietitians (RDs) with hands-on experience in dietary care of CKD patients, i.e., Ms. Rebecca Schlueter, Ms. Joanne Cooke, Ms. Amanda Brown-Tortorici, Ms. Meghan Donnelly, and Ms. Sherry Schulman. In response to these comments, they have reminded us that when they completed their registered dietitian board exam certification in renal nutrition (recertification of which is required every 5 years by the RD’s through the “Academy of Nutrition and Dietetics”, AND), they require to understand and manage the “results of urine chemistries related to metabolic status” including the expectations to apply 24-hour urine results for sodium, potassium, urea, creatinine, oxalate and uric acid, among others, as important parts of nutritional assessment.
With regard to the telehealth, the Veterans Affairs (VA) system in the USA have engaged VA dietitians with video connect and phone connect (VVC) appointments for a few years already, and the use of telehealth by dietitians have been enhanced under the recent unfortunate Covid-19 pandemic. Non-VA/public application of video telehealth expanded under the CMS 1135 waiver and Coronavirus Preparedness and Response Supplemental Appropriations Act. This includes Registered Dietitians (RDs) as providers of Medical Nutrition Therapy (MNT) see: https://www.cms.gov/newsroom/fact-sheets/medicare-telemedicine-health-care-provider-fact-sheet To address this important point, we have added the following to the revised version, from Line 303:
Registered dietitian who specialize in the field of renal nutrition are usually trained to use the 24-hour urine data, which may have an impact on accurate interpretation of daily nutrient intake estimates and assessment of patients adherence to the recommended medical nutrition therapy.6 Both dietitians and other healthcare providers use telehealth increasingly frequently since the inception of COVID-19 pandemic.7
- -Although authors have explained about the implementation of this diet, it is still unclear if it is possible for a CKD patient to really use this diet in long-term. For example, in most cases, budget is one of the important factors which dominants the dietary intake. In general, plant-based diets can be more expensive than a typical Western diet. Please explain how the socioeconomic status of CKD patients (and their families) will be taken into consideration when it comes to PLADO regiment.
OUR RESPONSE: We appreciate this important comment of Reviewer 2. Evidence suggests that a plant-based diet costs less than $4/day, while a meat-based traditional diet in the USA costs $12/day as show here by the UC Davis Integrative Medicine:
https://ucdintegrativemedicine.com/2015/03/cheap-or-expensive-the-real-truth-about-plant-based-diets/#gs.8hxovh
It is interesting to know that according to the paper by Fulgoni and Drewnowski based on the data from the US National Health and Nutrition Examination Survey (NHANES) 2013-2014 (Front Nutr. 2019;6:37),8 every commensurate plant-based food option is less expensive than animal based food set including meat and poultry, including in terms of cost per serving. Whole grains, legumes, and nuts/seeds are also cheaper in cost per unit of energy (calorie). These data are even more important knowing the subsidies that often go into the meat industry. Moreover, the same publication shows that compared to what racial/ethnic minorities typically spend per day on food ("NHANES" column, under $6), the only USDA healthy eating pattern than would come close is the vegetarian one. Additionally, whereas the recommended plant-based foods have relatively stable market value, we would like to clarify that manufacturers may profit by making processed pseudo-vegan foods, but these foods are not what we strive to promote under the CKD diet. Some of these discussions are specific to the field of dietetics and beyond the scope of our article. We have added the following, starting Line 465:
Indeed, whole food plant-based diet can help reduce weight in overweight and obese persons and help improve lipid profile and other risk factors related to cardiovascular disease or diabetes,3 and they are also more cost-effective than meat-dominant foods.5
Once again, we appreciate very much the evaluation and suggestions by Reviewer 2.
References for this response letter:
- Joshi S, Ostfeld RJ and McMacken M. The Ketogenic Diet for Obesity and Diabetes-Enthusiasm Outpaces Evidence. JAMA internal medicine. 2019;179(9):1163-1164. doi: 10.1001/jamainternmed.2019.2633. PubMed PMID: 31305866. URL: https://www.ncbi.nlm.nih.gov/pubmed/31305866.
- McMacken M and Shah S. A plant-based diet for the prevention and treatment of type 2 diabetes. J Geriatr Cardiol. 2017;14(5):342-354. doi: 10.11909/j.issn.1671-5411.2017.05.009. PubMed PMID: 28630614; PMCID: PMC5466941. URL: https://www.ncbi.nlm.nih.gov/pubmed/28630614.
- Wright N, Wilson L, Smith M, Duncan B and McHugh P. The BROAD study: A randomised controlled trial using a whole food plant-based diet in the community for obesity, ischaemic heart disease or diabetes. Nutr Diabetes. 2017;7(3):e256. doi: 10.1038/nutd.2017.3. PubMed PMID: 28319109; PMCID: PMC5380896. URL: https://www.ncbi.nlm.nih.gov/pubmed/28319109.
- Wang DD and Hu FB. Precision nutrition for prevention and management of type 2 diabetes. The lancet Diabetes & endocrinology. 2018;6(5):416-426. doi: 10.1016/S2213-8587(18)30037-8. PubMed PMID: 29433995. URL: https://www.ncbi.nlm.nih.gov/pubmed/29433995.
- Tuso PJ, Ismail MH, Ha BP and Bartolotto C. Nutritional update for physicians: plant-based diets. Perm J. 2013;17(2):61-6. doi: 10.7812/TPP/12-085. PubMed PMID: 23704846; PMCID: PMC3662288. URL: https://www.ncbi.nlm.nih.gov/pubmed/23704846.
- Sugimoto M, Asakura K, Masayasu S and Sasaki S. Relationship of nutrition knowledge and self-reported dietary behaviors with urinary excretion of sodium and potassium: comparison between dietitians and nondietitians. Nutr Res. 2016;36(5):440-51. doi: 10.1016/j.nutres.2015.12.012. PubMed PMID: 27101762. URL: https://www.ncbi.nlm.nih.gov/pubmed/27101762.
- Kalantar-Zadeh K and Moore LW. Impact of Nutrition and Diet on COVID-19 Infection and Implications for Kidney Health and Kidney Disease Management. J Ren Nutr. 2020;30(3):179-181. doi: 10.1053/j.jrn.2020.03.006. PubMed PMID: 32291198; PMCID: PMC7186539. URL: https://www.ncbi.nlm.nih.gov/pubmed/32291198.
- Fulgoni V, 3rd and Drewnowski A. An Economic Gap Between the Recommended Healthy Food Patterns and Existing Diets of Minority Groups in the US National Health and Nutrition Examination Survey 2013-14. Front Nutr. 2019;6:37. doi: 10.3389/fnut.2019.00037. PubMed PMID: 31019912; PMCID: PMC6458255. URL: https://www.ncbi.nlm.nih.gov/pubmed/31019912.